# Mathematical Modeling of *Salmonella* Cancer Therapies Demonstrates the Necessity of Both Bacterial Cytotoxicity and Immune Activation

**DOI:** 10.3390/bioengineering12070751

**Published:** 2025-07-10

**Authors:** Lars M. Howell, Neil S. Forbes

**Affiliations:** 1Department of Chemical Engineering, University of Massachusetts, Amherst, MA 01003, USA; larshowell@umass.edu; 2Department of Microbiology, University of Massachusetts, Amherst, MA 01003, USA; 3Institute for Applied Life Sciences, University of Massachusetts, Amherst, MA 01003, USA

**Keywords:** cancer, *Salmonella* therapies, ordinary differential equation model, immunoengineering, T cell cytotoxicity

## Abstract

*Salmonella* therapies are a promising tool for the treatment of solid tumors. *Salmonella* can be engineered to increase their tumor infiltration, cell killing abilities, and immunostimulatory properties. However, bacterial therapies have often failed in clinical trials due to poor characterization. Mathematical models are useful for predicting the immune response to cancer treatments and characterizing the properties of bacterial invasion. Herein we develop an ordinary differential equation-based model that combines bacterial therapies with classical anti-tumor immunotherapies. Our modeling results suggest that increasing bacterial localization to the tumor is key for therapeutic efficacy; however, increased intracellular invasion and direct bacterial mediated cytotoxicity does not reduce tumor growth. Further, the model suggests that enhancing T cell-mediated cell death by both bacterial stimulation of pro-inflammatory cytokines and activation of T cells via antigen cascade is critical for therapeutic efficacy. A balance of intracellular and extracellular *Salmonella* leads to more effective therapeutic response, which suggests a strategy for strain design to be tested in vivo. Overall, this model provides a system to predict which engineered features of *Salmonella* therapies lead to effective treatment outcomes.

## 1. Introduction

Bacterial therapies are a promising new cancer immunotherapy, with multiple clinical trials underway (NCT05038150, NCT06631079, NCT02625857). It is important to understand which factors are key in enhancing the efficacy of bacterial therapies and inducing tumor clearance. Engineered *Salmonella* for the treatment of solid tumors offers benefits including tumor targeting, delivery of cytotoxic molecules, and immune activation from both delivered molecules and innate bacterial properties [1,2]. *Salmonella* therapies are highly engineerable, allowing for precise control of tumor infiltration, cell invasion, and programmed cell death [3,4]. In murine models, bacterial therapies have wide variability, with some clearing tumors entirely while others result in minimal efficacy [5]. In human clinical trials, *Salmonella* therapies have not demonstrated significant efficacy [6]. Mathematical modeling has the potential to elucidate the mechanisms that regulate therapeutic responses in cancer immunotherapy.

Currently, bacterial cancer therapies are thought to operate via two pathways. The first is by direct tumor cell cytotoxicity and the second is by redirecting the immune response towards cancerous cells [1,7]. *Salmonella* engineered to deliver antigens redirect T cell cytotoxicity towards tumor cells [5,8,9,10,11]. For example, the intracellular delivery of exogenous antigens to cancer cells by *Salmonella* results in tumor reduction by antigen-specific T cells generated from a prior vaccination [5]. Bacterial systems of antigen delivery have been used to educate immune cells to recognize tumor-associated antigens present in patient-specific tumors [12]. Intracellular bacteria can also result in the cancer cell presentation of bacterial epitopes, which are targets for immunogenic cell death [11,13,14]. These systems rely on T cells that specifically target invaded cancer cells.

*Salmonella* therapies are highly tumor specific and have direct cytotoxic mechanisms. *Salmonella* clear rapidly from the blood while remaining in tumors at ratios of 1000:1 to healthy tissue [6,7,15,16]. Tumor specificity can be enhanced by engineering *Salmonella* to be auxotrophic to amino acids [17] or increasing their motility by increasing flagellar presentation [4]. *Salmonella* are naturally intracellular pathogens, which allows for direct interference with cellular mechanisms. *Salmonella* invade cells using the type III secretion system (T3SS), which is dependent on physical contact with cancer cells [18]. Once inside cells, engineered *Salmonella* can deliver proteins to the cytosol, including toxic payloads such as caspase-3 [3] and Cp53 [19]. The rate of intracellular invasion is increased by upregulating flagella [4] and modifying *Salmonella* pathogenicity island (SPI) protein expression [20].

Direct cell cytotoxicity is insufficient for effective bacterial therapies. Increasing the immunogenic tumor microenvironment is essential for tumor clearance. Recognition of *Salmonella* pathogen-associated molecular patterns (PAMPs) activates innate immune cells via Toll-like receptors (TLRs) and leads to the production of cytokines, including TNFα, IFNγ, and IL-6, which convert the immune microenvironment from cold to hot [21,22,23,24]. *Salmonella* have also been engineered to directly deliver immunostimulatory cytokines [25,26,27]. By inducing a strong pro-inflammatory response, *Salmonella* function as adjuvants, increasing the efficacy of T cell activation and anti-tumor immunity [28].

Anti-tumor immunity is critical for increasing therapeutic efficacy. Immunogenically, hot tumors have high levels of activated T cells and pro-inflammatory cytokines, and respond well to immunotherapies [29]. Cancer cell death releases tumor associated antigens (TAAs) into the microenvironment, which are picked up by antigen presenting cells (APCs) and presented to naïve T cells, educating them to that target and generating broad anti-tumor immunity [30]. Once T cells have been educated to a specific epitope, they can re-activate in response to its detection without undergoing this education process and triggering rapid immunogenic cell death [31].

Mathematical models of cancer are critical to understanding the mechanisms undergirding immunotherapies and have been used to design more effective therapies. Mathematical models describing the relationship between cancer cells and immune cells have shown that T cells are a critical cell type in controlling tumor growth [32,33,34,35]. Partial differential equation (PDE) models describing bacterial invasion into tumors showed that chemotactic affinity to tumors drives bacterial accumulation. These models, however, did not explore the immune impact of bacterial therapies [4,36,37]. Models of Bacillus Calmette–Guerin (BCG) in bladder cancer, the only approved bacterial therapy, showed that increasing effector T cell densities reduced the growth of bacterially infected cancer cells [38,39]. This study, however, did not explore how bacteria could be engineered to improve therapeutic outcomes [38,39]. Models of oncolytic virus therapies showed that anti-viral and anti-tumor immunity together were critical for controlling tumor progression [40,41]. Oncolytic viruses are similar to bacterial therapies in that they both rely on invading cancer cells to initiate cell death and immune targeting. However, the established models of oncolytic viruses cannot be directly translated to bacterial therapies because of their physiological differences, including their viable extracellular form, active invasion, and self-contained translation machinery.

In this work, we developed an ordinary differential equation (ODE) model of antigen delivering *Salmonella* therapies to explore the critical parameters for effective bacterial therapies including invasion rate, immunostimulatory activity, and direct cytotoxicity. The model determined relationships between the extra- or intracellular location of *Salmonella* and the resulting cytokine production and T cell activation. Further, we modeled the impact of combination therapies, such as checkpoint blockade, by reducing the rate of T cell exhaustion. Overall, this model highlights the impact of modulating *Salmonella* rate kinetics to characterize the immune response to bacterial therapies. This provides a starting point for the more thoughtful engineering of anti-cancer bacterial treatments.

## 2. Materials and Methods

### 2.1. Model Assumptions

The model consists of eight ODEs that describe the interactions of tumor and immune cells after the administration of therapeutic bacteria (Figure 1). In the model, *Salmonella* are in two compartments: the blood stream and a tumor. *Salmonella* in tumors, *S_T_*, are dependent on colonization from the bolus injected in the blood, *S_B_*. Tumors are composed of cancer cells and immune cells. Initially, cancer cells do not contain bacteria. After bacterial invasion, they are recognized by exogenous antigen-specific T cells. The T cells in the model are divided into those that recognize antigens delivered by *Salmonella*, and those that are trained to recognize tumor antigens more broadly. Innate immune cells such as macrophages produce pro-inflammatory cytokines such as IFNγ and IL-6 in response to intratumoral *Salmonella*. The model system is run in MATLAB (MathWorks, R2023a) using the solver ode15s.

### 2.2. Model Description

#### 2.2.1. Bacterial Dynamics

*Salmonella* are injected into the blood, S_B_, before infiltrating into the tumor, *S_T_* (Equations (1) and (2)).(1)$$\frac{{dS}_{B}}{dt}=-k_{1}S_{B}$$(2)$$\frac{{dS}_{T}}{dt}=k_{1}S_{B}-k_{2}S_{T}T_{U}-d_{1}S_{T}$$

*Salmonella* are injected into the blood, *S_B_*, as a bolus at time *t* = 0 to begin treatment modeling, *S_B_*_,0_. *Salmonella* are cleared from the blood and infiltrate the tumor at the rate *k*_1_*S_B_* (Equations (1) and (2)). Additional *Salmonella* clearance without entering the tumor does not impact tumor cell proliferation (Appendix A). Once inside the tumor, *Salmonella*, S_T_, are trapped by the fenestrated tumor vasculature and neutrophils, preventing re-entry to the blood [42]. *Salmonella* invade cancer cells at the rate *k*_2_*S_T_T_U_* (Equations (1), (3) and (4)), which is dependent on the quantity of bacteria, *S_T_*, and uninvaded cancer cells T_U_. Bacteria death, caused by phagocytosis by intratumoral macrophages or NETosis by neutrophils, reduces the density of extracellular *Salmonella*, *d*_1_*S_T_* (Equation (2)). The population of intratumoral extracellular *Salmonella* is controlled by tumor infiltration, *k*_1_*S_B_*, cell invasion, *k*_2_*S_T_T_U_*, and bacterial death, *d*_1_*S_T_*.

#### 2.2.2. Tumor Cell Dynamics

Tumors contain both uninvaded and invaded cancer cells, whose populations are given by *T_U_* and *T_I_*, respectively (Equations (3) and (4)).(3)$$\frac{{dT}_{U}}{dt}=aT_{U}\left(1-b(T_{U}+T_{I})\right)-k_{2}S_{T}T_{U}-\frac{s_{1}T_{U}E_{T}}{h_{1}+E_{T}}$$(4)$$\frac{dT_{I}}{dt}=k_{2}S_{T}T_{U}-k_{3}T_{I}-\frac{s_{1}T_{I}E_{T}}{h_{1}+E_{T}}-\frac{s_{2}T_{I}E_{B}}{h_{1}+E_{B}}$$

Tumors grow following a logistic density growth rate before reaching their carrying capacity, which is defined by the total tumor volume, including both invaded and uninvaded tumor cells [32,43]. Tumor growth is defined as *aT_U_* (1 *− b (T_U_ + T_I_))* (Equation (3)), where a is the tumor growth rate and b is the inverse of the carrying capacity. Tumor volume is assumed to be proportional to the number of live tumor cells, *(T_U_ + T_I_).* Tumor growth is the change in tumor cell number over time. Once tumor cells have been invaded by *Salmonella* they do not proliferate at a significant rate [44]. Once *Salmonella* are intracellular, they can trigger cell death by releasing effector proteins or cytotoxic payloads [45,46]. Bacterial invasion causes cell death at given rate, *k*_3_*T_I_* (Equation (4)) [47].

Cancer cells in tumors are killed by interactions of cytotoxic CD8^+^ T cells. There are two pools of effector T cells, *E_B_* and *E_T_.* Bacteria-recognizing T cells, *E_B_*, are memory T cells that only target bacterially invaded cancer cells by recognition of the delivered exogenous antigen. The other type of effector T cells are pan-tumor recognizing T cells; these T cells recognize tumor-associated antigens (TAAs), which are self-peptides presented by cancer cells that have mutated beyond the healthy range to be recognized as foreign by the immune system [30,48]. Immunogenic cell death is dependent on the structure of the tumor microenvironment [49] and the ratio of T cells (*E_T_* or *E_B_*) and cancer cells (*T_U_* or *T_I_*). It saturates at high densities of T cells, *sT*. At sub-saturation densities of T cells, immunogenic cell death, $\frac{sTE}{h_{1}+E}$ (Equations (3) and (4)), is dependent on saturation parameter, h_1_, which defines the quantity of immune cells for the half-maximal killing rate.

#### 2.2.3. Immune Cell Dynamics

T cells are activated by exposure to immunogenic epitopes (Equations (5)–(7)).(5)$$\frac{dE_{B}}{dt}=\frac{t_{1}E_{N}T_{I}}{h_{2}+T_{I}}\frac{I_{C}}{g_{1}+I_{C}}-d_{2}E_{B}T_{I}-d_{3}E_{B}$$(6)$$\frac{dE_{T}}{dt}=\frac{t_{2}E_{N}\left(T_{U}+T_{I}\right)}{h_{3}+T_{U}+T_{I}}\frac{I_{C}}{g_{1}+I_{C}}-d_{2}E_{T}\left(T_{U}+T_{I}\right)-d_{3}E_{T}$$(7)$$\frac{dE_{N}}{dt}=i-\frac{t_{1}E_{N}T_{I}}{h_{2}+T_{I}}-\frac{t_{2}E_{N}\left(T_{U}+T_{I}\right)}{h_{3}+T_{U}+T_{I}}$$

T cell activation is driven by contact with tumor cells expressing foreign epitopes. In vivo, this process is driven by dendritic cell travel to lymph nodes and the activation of naïve T cells from picked up antigens—for simplicity, this activation process and travel has been folded into the activation rates *t*_1_ and *t*_2_. Similar to immunogenic cell death, T cell activation is limited by an upper threshold [50]. At high densities of antigen-presenting tumor cells, activation is saturated, *tE*. At sub-saturation levels, T cell activation, $\frac{tET}{h+T}$ (Equations (5)–(7)), is dependent on the saturation parameter h_2_ or h_3_. T cell activation is also dependent on the presence of pro-inflammatory cytokines such as TNFα and IFNγ until they reach the saturation threshold, *g*_1_. Below this threshold, T cell activation is limited by $\frac{I_{C}}{g_{1}+I_{C}}$ (Equations (5) and (6)). All T cells are activated from naïve T cells, E_N_. The pool of naïve T cells is maintained by the spleen and flux of new cells into the tumor, i (Equation (7)) [51]. T cell exhaustion and cell death is described by the parameters d_2_ and d_3_ (Equations (5) and (6)). The term d_2_ describes cell exhaustion after contact with tumor cells and d_3_ marks the half-life of activated T cells. The activated T cell dynamics are expressed in Equations (5) and (6).

#### 2.2.4. Cytokine Dynamics

In tumors, cytokines are produced in response to bacteria (Equation (8)).(8)$$\frac{dI_{C}}{dt}=l_{1}S_{T}-l_{2}I_{C}$$

Innate immune cells within the tumor, such as macrophages, detect *Salmonella* and produce inflammatory cytokines in response [52]. *Salmonella* can also be engineered to release these cytokines [25,53]. Intratumoral *Salmonella* induce the production of cytokines at rate *l*_1_*S_T_* (Equation (8)). Cytokines degrade over time at *l*_2_*I_C_* (Equation (8)). Together these assumptions generate a set of eight ODEs that govern the tumor response to *Salmonella* therapies: Equations (1)–(8).

*Salmonella* injection into the blood is set to *S_B_*_,0_ = 10^6^ CFU based on experimental values used in the lab [22]. It is assumed that before treatment there would be no *Salmonella* in the blood, *S_T_*_,0_ = 0 CFU. Tumor size at treatment initiation is set to *T_U_*_,0_ = 10^6^ cells. Just as there is assumed to be no *Salmonella* initially in the tumor, we assume that there are no invaded cancer cells at *t* = 0, thus *T_I_*_,0_ = 0 cells. We assume that there is a small number of naïve T cells in the tumor at *t* = 0, and set *E_N_*_,0_ = 100 cells. We set that there are no activated T cells at the initiation of therapy: *E_B_*_,0_ = 0 cells, *E_T_*_,0_ = 0 cells. The quantity of initial cytokines is dependent on the immunological state “hot” or “cold” of the tumor and is discussed in detail in the results, and, in the majority of the experiments, is set to *I_C_*_,0_ = 10 ng/mL [54].

### 2.3. Parameter Estimations

Parameters are estimated to capture the relationships between the cell types in the tumor microenvironment (Table 1). Parameters are selected from a mixture of in vitro, in vivo, and modeling papers, which each provide limitations on their accuracy in the heterogeneous tumor microenvironment. Rates of *Salmonella* tumor infiltration and cellular invasion are calculated from a mixture of in vitro and in vivo data. Administration of attenuated *Salmonella* in human clinical trials demonstrates rapid clearance from the blood, k_1_, at a rate of 0.5 (day^−1^) [6]. Tumor-on-a-chip microfluidic devices from Raman et al. [4]. are used to characterize intracellular invasion, k_2_, of *Salmonella* and it is found that bacteria invade cells at a rate of 2.97 × 10^−6^ cells day^−1^, though this rate is highly engineerable by modulating the expression of flagella or T3SS proteins. Microfluidic devices in vitro are used to characterize bacterially mediated cell death; data from Toley and Forbes [47] is used to calculate the rate constant k_3_ as 3.20 day^−1^. *Salmonella* do not remain in tumors indefinitely. The clearance rate of *Salmonella*, d_1_, is considered to be a combination of effects, including phagocytosis by macrophages or other immune cells [55]. The rate constant is estimated to be 0.001 day^−1^.

Tumor growth parameters have been highly characterized by immune system models. Cancer cells proliferate, a, at a rate of 0.43 day^−1^ until they reach a carrying capacity of 9.80 × 10^8^ cells, which is 1/b. Both values are acquired from the modeling work of de Pillis et al., which calculates rate constants based on the experimental work of Diefenbach et al. [32,56,57]. Parameters are selected from commonly cited cancer immunotherapy models.

The activation rate of naïve, t_2_, and memory, t_1_, T cells is determined using data from previous studies, including Zimmermann et al., which analyzes the rate of presentation of CD69, CD25, and the T cell lysis ability overtime in vitro [58]. Reactivation of memory T cells occurs at t_1_ = 1.50 days^−1^, while naïve T cells activate and differentiate at t_2_ = 1.23 days^−1^. Data analyzing the activation of T cells from melanoma patients ex vivo is used to confirm activation timelines [59,60]. Finally, calculated values are compared with other values used in the immune modeling literature [32,33,34,35,40]. The values for the half-maximal activation rate point, h_2_ = h_3_ = 2.50 × 10^3^ cells, are taken from prior T cell cancer immunotherapy modeling work [61,62].

The rate of T cell lysis of cancer cells, s_1_ = 2.87 day^−1^ and s_2_ = 2.13 day^−1^, is determined by analyzing the kinetics of target cell lysis by T cells at various effector to target cell ratios in vitro [49,63]. There is found to be broad diversity between highly cytotoxic individual T cells and less toxic T cells, demonstrating that this rate can be highly variable [49,63]. These values are also compared with the lysis rate presented in other modeling literature [32,40]. The saturation constant h_1_ is determined from prior cancer immunotherapy modeling [33].

The rate constant of T cell exhaustion, d_3_, is determined by the known half-life of T cells [33,64] and is calculated as d_3_ = ln(2)/half-life_Tcell_ = 0.11 days^−1^. The influx rate of naïve T cells, i, is calculated from population data analyzing thymic output of naïve T cells in humans [65]. The rate of T cell death after interaction with cancer cells, d_2_ = 3.03 × 10^−10^ cells^−1^ day^−1^, is acquired from other modeling literature [57,61,66].

The rate of cytokine production, l_1_ = 0.51 day^−1^, by macrophages in response to *Salmonella* is measured using a study that invaded macrophages with *Salmonella* for 30 min and then calculated the amounts of TNFα, IL-6, and IL-12 produced 24 h later [67]. This value is also compared with the production of IFNγ in murine tumor models after treatment with hCXCL16 [68], and the rate of secretion of *Salmonella* effectors after invading macrophages [69].

The saturation constant, g_1_ = 120 ng mL^−1^, is calculated using IL-2 as an example cytokine and examining the dose–response curve of pSTAT5 in vitro [70,71]. The impact of IL-2 injection in vivo on T cell proliferation is examined and the maximal effective dose is utilized [72]. Cytokine sensing, such as for IFNγ, can be suppressed by tumor cells by the expression of surface molecules such as PD-L1 of the secretion of other cytokines such as TGFβ [73]. In rats, the degradation rate, l_2_, of IL-6 is calculated after systemic injection of a cytokine bolus and measuring the serum concentration over four days [74]. Cytokine degradation is also compared with other modeling studies.

**Table 1 bioengineering-12-00751-t001:** Model parameters, units, and sources.

Parameter	Description	Units	Starting Value	Sources
k_1_	Bacterial tumor invasion rate	day^−1^	0.5	[6]
k_2_	Bacterial cell invasion rate	cell^−1^ day^−1^	2.97 × 10^−6^	[4]
d_1_	Bacterial half-life	day^−1^	0.001	[55]
a	Tumor growth rate	day^−1^	0.43	[32,56,57]
b	Tumor carrying capacity	cells^−1^	1.02 × 10^−9^	[32,56]
s_1_	Tumor epitope specific T cell killing rate of cancer cells	day^−1^	2.87	[32,40,49,63]
h_1_	Half-max concentration for T cell to cancer cell killing	cells	5 × 10^3^	[33]
k_3_	Bacteria-based tumor cell death	day^−1^	3.20	[47]
s_2_	Bacterial antigen-specific T cell rate of killing invaded cells	day^−1^	2.13	[32,40,49,63]
t_1_	Rate of bacterial antigen memory T cell activation	day^−1^	1.50	[58,59,60,61]
h_2_	Half-max conc for bacterial antigen memory T cell activation	cells	2.50 × 10^3^	[61,62]
d_2_	Tumor killing of immune cells	cell^−1^ day^−1^	3.03 × 10^−10^	[57,61,66]
d_3_	Immune cell half-life	day^−1^	0.11	[33,64]
t_2_	Rate of pan-tumor naïve T cell activation	day^−1^	1.23	[58,59,60,61]
h_3_	Half-max conc for pan-tumor naïve T cell activation	cells	2.50 × 10^3^	[61]
i	Source of naïve T cells	cells day^−1^	90	[65]
l_1_	Production of cytokines in response to bacteria	ng mL^−1^ cell^−1^ day^−1^	2.88–5.76	[67,68,69]
l_2_	Degradation of cytokines	day^−1^	0.51	[35,74]
g_1_	Half-max cytokine conc for T cell activation	ng mL^−1^	120	[32,35,70,71,72]

### 2.4. Sensitivity Analysis

Sensitivity analysis is performed using Latin hypercube sampling/partial rank correlation coefficient (PRCC) [75]. Five hundred samples are run and used to calculate the partial rank correlation coefficients (PRCCs) of the rate constants. Briefly, each parameter of the model is randomly assigned a value within an order of magnitude of the literature determined values. The model is run using these parameters. The total tumor cell number, *T_U_ + T_I_*, 15 days after bacterial injection is plotted against the randomly generated values for each parameter to determine their impact on tumor cell count. For tumor growth rate, a, and carrying capacity, b, parameters are chosen within a factor of two as these values are more influential than other factors on final tumor cell count. Sensitivity analysis is also performed on a bacteria free system to determine which variables are not specific to bacterial immunotherapies but rather immunotherapies more broadly.

Sensitivity analysis shows that the rates of bacterial infiltration into tumors and intracellular invasion, alongside the T cell activation rate and tumor growth rate, are the parameters that impact the final tumor cell number the most. When no *Salmonella* are injected into the blood stream (Figure 2A), the model is dominated by the tumor growth rate a (*p* < 10^−60^). Other significant factors include the half-maximum density for cytokine-based T cell activation g_1_ (*p* = 0.000244), the influx rate of naïve T cells i (*p* = 0.038601), and the half-maximum density for T cell-based killing of tumor cells h_1_ (*p* = 0.00010055). However, these factors have minimal impact compared with tumor growth rate. More available T cells in the tumor, from influx, i, reduce the tumor volume as there is some activation from exposure to tumor neoantigens. The amount of this activation is based on how many naïve T cells there are to be educated. Increasing the saturation limits for cytokine T cell activation and T cell lysis of tumor cells increases tumor growth as they reduce cell activation or tumor cell lysis, respectively, by increasing the amount of cytokines or T cells that must be present to reach the maximal rate.

When 10^6^
*Salmonella* are injected at time t = 0 (Figure 2B), there is a greater variability in the key parameters. The rates of *Salmonella* invasion into the tumor k_1_ (*p* = 3.6213 × 10^−60^) and cancer cells k_2_ (*p* = 3.094 × 10^−36^) are negatively correlated with the tumor cell count. This suggests that these parameters would be key in engineering more effective bacterial therapies. As in the bacteria free system, the tumor growth rate a (*p* = 1.568 × 10^−50^), the influx of naïve T cells i (*p* = 1.0094 × 10^−8^), and the half-maximum density for T cell-based killing of tumor cells h_1_ (*p* = 1.563 × 10^−22^) significantly impact tumor growth. As tumor growth, a, is no longer overwhelmingly dominant in impacting tumor volume, the relative impact of other terms increases. Additionally, T cell half-life d_3_ (*p* = 0.0087176), the rate of activation of pan-tumor T cells t_2_ (*p* = 8.7671 × 10^−7^), and the rate of pan-tumor T cell killing of cancer cells s_1_ (*p* = 3.7209 × 10^−18^) significantly impact tumor volume.

## 3. Results

### 3.1. The Route of Bacterial Administration Impacts Tumor Growth

The route of bacterial injection impacts intracellular accumulation and tumor volume over time. The direct injection of *Salmonella* into tumors can increase therapeutic efficacy; however, it relies on larger more accessible tumors. We modeled the difference between intravenous and intratumoral bacterial treatments by modifying the initial conditions of blood-born, *S_B_*, and intratumoral, *S_T_*, bacteria (Figure 3). For an intravenous injection, *S_B_*_,0_ was set to 10^6^ CFU, which mimics an intravenous bolus injection of bacteria. Immediately after an injection, there are no *Salmonella* in the tumors and *Salmonella* invade over time. Intratumoral injection was modeled using the initial conditions *S_B_*_,0_ = 0 and *S_T_*_,0_ = 10^6^ CFU, which bypasses the tumor infiltration process. Intratumoral injections of bacteria reduce tumor growth more than intravenous injections (Figure 3A). Intravenous injection results in slower invasion of cancer cells than intratumoral injection as the bacteria must first infiltrate the tumor from the blood (Figure 3B). However, intravenously *Salmonella* injected have a longer duration of remaining inside the invaded cancer cells due to *Salmonella* entering the tumor microenvironment from the blood over the first five days (Figure 3B). Overall, both methods of bacterial injection result in similar numbers of invaded cancer cells during the first 10 days. The area under the curve in intravenously injected *Salmonella* is 2.9963 × 10^5^, while that of intratumorally injected *Salmonella* is 3.0694 × 10^5^, indicating that it is the timing of cell invasion rather than the total number of invaded cells that impacts efficacy (Figure 3B).

Intratumorally injected bacteria rapidly invade cancer cells, with a peak invaded cell number of 2.4 × 10^5^ tumor cells, seven hours post-injection (Figure 3C). The increased population of invaded tumor cells, and the corollary decrease in uninvaded tumor cells has a significant effect on the overall tumor volume. Invaded cells do not proliferate and are targets for T cells that recognize both bacterially delivered antigens, as well as pan-tumor recognizing T cells. Intravenous *Salmonella* injection results in a small fraction of cancer cells being invaded with bacteria (Figure 3D). After intravenous injection, *Salmonella* clear rapidly from the blood, with fewer than one tenth of the original bacterial number remaining after 5 days (Figure 3E). Just over a tenth of the initial density of bacteria are present in the tumor microenvironment one day after injection, reaching a maximal intratumoral density of 1.1 × 10^5^ CFU of *Salmonella* (Figure 3F). With this initial dose of bacteria and tumor volume, fewer than one tenth of the total volume of cancer cells are invaded by *Salmonella* (Figure 3D). Peak intracellular *Salmonella* density, *T_I,max_*, occurs one day after bacterial injection, highlighting the rapid invasion of *Salmonella* after treatment. The large proportion of cancer cells remain uninvaded by *Salmonella*, indicating that direct *Salmonella*-mediated cell death is not the major cause of tumor suppression in this system.

Intratumoral and intravenous injections have opposite effects on the activation of the two cohorts of T cells. Intravenous injection activates more bacterial antigen-specific T cells than intratumoral injection, as there is a longer tail in the presence of invaded cancer cells, which can activate bacteria-specific T cells (Figure 3G). This trend is reversed for pan-antigen T cells (Figure 3H), because more bacteria in the tumors increases cytokines and draws more T cells in general into the tumor. Continued exposure to bacterially invaded cancer cells results in greater activation of bacterial antigen-specific T cells.

### 3.2. Characterization of Hot and Cold Tumor Models

Increasing cytokine levels creates immunogenically hot tumors that have reduced tumor cell proliferation and increased T cell activation compared with cold tumors (Figure 4). The immunological state of the tumor is modeled by adjusting the initial cytokine concentration and the cytokine degradation rate. Immunologically cold tumors have minimal inflammatory cytokines, which clear rapidly when they are produced (*I_C_*_,0_ = 10 ng/mL and *l*_2_ = 0.5137 day^−1^) [54]. Immunogenically hot tumors have significantly higher initial cytokine levels [29], which are constantly present in the tumor microenvironment (*I_C_*_,0_ = 100 ng/mL and *l*_2_ = 0 day^−1^) [54]. Immunosuppressed (cold) and immune-active (hot) tumors grow at different rates, even in the absence of bacterial treatment (Figure 4A). In cold tumors, maximal tumor volume is reached 22 days after tumor formation (*T_U_*_,0_ = 10^6^ cells; Figure 4A). The reduction in tumor cell number in hot tumors is driven by an increase in pan-tumor cytotoxic T cells activated by the interactions of naïve T cells with tumor cells (Figure 4B). There is minimal T cell activation in untreated cold tumors (Figure 4B). These results match what has been seen clinically, where patients with highly immune active tumors and many activated T cells have tumor remission or increased survival compared with patients with no T cell activation [76].

Treatment with *Salmonella* delays tumor cell proliferation in both hot and cold tumors compared with untreated tumors, but cold tumors overcome the initial treatment efficacy (Figure 4C,D). Bacterial injection (*S_B_*_,0_ = 10^6^ CFU at t = 0) delays tumor growth in cold tumors for 20 days, after which, the tumors grow exponentially (Figure 4C). In hot tumors, *Salmonella* treatment slows tumor growth through day 30 (Figure 4C). In both tumors there is rapid proliferation of pan-tumor T cells through day 17 (Figure 4D). In cold tumors, these cells begin to decrease after day 17, resulting in increased tumor volume (Figure 4D). In hot tumors, pan-tumor T cells continue increasing (Figure 4D). Transforming tumor microenvironments from cold to hot is one feature of bacterial therapies [22,23,24]. Moving forward, we analyze the response to bacterial therapies in immunologically cold tumors (*I_C_*_,0_ = 10 ng/mL and *l*_2_ = 0.5137 day^−1^), as this is the case where bacterial therapies are more likely to be clinically applied.

### 3.3. Tumor Volume and Bacterial Injection Density

Increasing the quantity of delivered bacteria reduces tumor growth and increases T cell activation, while increasing the initial tumor volume reduces therapeutic efficacy (Figure 5). With a small initial tumor volume (*T_U_*_,0_ = 10^5^ cells), an injection of 10^6^
*Salmonella* is sufficient to completely clear the tumor (purple) and 10^4^
*Salmonella* induces significant growth delay (yellow; Figure 5A). With a larger tumor (*T_U_*_,0_ = 10^6^ cells), bacterial treatments up to 10^6^
*Salmonella* do not completely regress the tumor but cause significant growth delay (Figure 5D). Large tumors at the time of treatment (*T_U_*_,0_ = 10^7^ cells) result in a delay of tumor growth; however, all tumors reach the carrying capacity by 30 days post-injection (Figure 5G).

The activation of bacterial antigen-responsive T cells (Figure 5B,E,H) is proportional to the amount of *Salmonella* delivered, with higher initial doses of *Salmonella* leading to more bacteria-specific T cells. Doses above 10^4^ CFU of *Salmonella* are key for significant bacterial T cell activation and efficacy. Bacterial antigen-specific T cells (*E_B_*) peak more rapidly than pan-tumor T cells (*E_C_*), with *E_B_* peaking between 5 and 10 days depending on initial tumor volume (Figure 5B,E,H) and *E_C_* peaking between 5 and 20 days depending on initial volume (Figure 5C,F,I). This delay occurs because T cells responding to bacterial antigens are memory T cells being reactivated, while pan-tumor T cells are naïve T cells which must undergo somatic recombination in response to antigen stimulation to activate.

In small tumors (10^5^ cells), there is a greater activation of pan-tumor T cells, with 10^4^ CFU (yellow) of *Salmonella* activating more T cells than 10^6^ CFU (purple; Figure 5C). This is due to the rapid reduction in tumor volume caused by the larger quantities of *Salmonella* and bacterial-specific T cells. Tumor clearance eliminates the need for continuous T cell activation and removes the source of activation stimulus. Fewer pan-tumor T cells from smaller initial quantities of *Salmonella* is a relationship that is only present when the tumor is cleared from treatment. With larger tumors at the time of treatment, more *Salmonella* result in more pan-tumor T cells (Figure 5F,I).

### 3.4. Engineering Salmonella to Improve Therapeutic Efficacy

*Salmonella* therapies are effective due to the high levels of control of the bacterial properties through genetic engineering. *Salmonella* have been engineered that specifically infiltrate tumors [17], invade cancer cells [4], and lyse within cancer cells [3]. However, the impacts of these changes have not been fully characterized towards therapeutic outcomes. We analyze the impact of each of these factors on tumor volume and T cell activation.

As intratumoral injections result in greater therapeutic efficacy (Figure 3), we hypothesize that increasing the rate of bacterial infiltration of the tumor would also improve efficacy (green, Figure 6A–D). Changing the rate of invasion into the tumor (*k*_1_) decreases tumor growth (Figure 6A). Increasing the rate of tumor invasion increases the peak number of invaded cancer cells, and moves the time of maximal invasion earlier (Figure 6B). Despite there being more invaded cancer cells, increasing tumor invasion decreases the overall number of bacterial antigen-responding T cells (Figure 6C). The sharp peak and rapid decrease in intracellular bacteria reduce the stimulation required for bacterial antigen T cell activation. A large peak in invaded cancer cells does not increase T cell activation indefinitely because the rate of activation plateaus, due to its maximal rate. With lower *k*_1_, the number of invaded cancer cells remains higher at later time points (Figure 6B). This pool of invaded cancer cells maintains the activation of bacterial targeting T cells throughout the experiment (Figure 6C). This trend is not seen with pan-tumor T cells, where increasing the tumor infiltration rate increases the overall number of pan-tumor T cells (Figure 6D).

We hypothesize that increasing cellular invasion would decrease tumor cell proliferation, as bacterially invaded cells do not proliferate and would be targetable by both bacteria-specific and pan-tumor T cells (blue, Figure 6E–J). However, changing the intracellular invasion rate has a more interesting impact on tumor growth. In the first ten days after bacteria injection, increasing the invasion rate reduces tumor growth, as expected (Figure 6F). However, as tumors continue to grow, the increased invasion rate results in larger tumors (Figure 6E). Slowing the invasion rate leads to both a delayed peak in invaded cancer cells at seven days rather than one day post-injection, and a decrease in the number of invaded cancer cells, with only 4 × 10^4^ cells being invaded rather than 10 × 10^4^ (Figure 6G). Increasing the invasion rate decreases the formation of both bacteria antigen-specific T cells and pan-tumor T cells (Figure 6H,I). Increasing the rate of intracellular invasion decreases the formation of pan-tumor T cells more in the transition from *k*_2_ = 10^−4^ to 10^−6^ than the transition from 10^−6^ to 10^−8^ (Figure 6I), which corresponds to tumor growth taking off in the case where *k*_2_ = 10^−4^ after day 15, while the *k*_2_ = 10^−6^ and 10^−8^ conditions remain contained (Figure 6E). Increasing the rate of intracellular invasion decreases the amount of cytokines produced in response to *Salmonella* (Figure 6J).

Increasing the rate of direct bacterial cytotoxicity decreases overall tumor growth (Figure 6K). Increasing bacterially mediated cell death decreases the numbers of invaded cancer cells overtime, as these cells die (Figure 6L). Increasing the rate of direct bacterial cytotoxicity to cancer cells, *k*_3_, decreases the population of bacterial T cells (Figure 6M) and increases that of pan-tumor T cells (Figure 6N). As bacteria directly lyse cancer cells, they present the delivered exogenous antigen for less time. This reduced presentation results in the formation of fewer anti-bacterial T cells. However, there is a still strong formation of pan-tumor T cells that are not dependent on invaded cancer cells.

### 3.5. Cytokine Production

Engineering bacteria to acutely produce cytokines, such as INF-γ, or increasing the bacterial presentation of PAMPs to increase the macrophage induction of TNFα reduces long-term tumor volume (Figure 7). The peak of cytokine production by bacteria occurs on day 3, which lightly trails the peak in intratumoral bacterial density (Figure 7A and Figure 2B). Increasing the rate of cytokine production decreases tumor volume, which is noticeable starting on day 15 (Figure 7B). There is no significant difference in the activation of T cells specific to exogenous antigens (Figure 7C). The rate of production of pan-tumor T cells remains consistent; however, increasing cytokine production increases the duration of T cell activation and the overall quantity of T cells that infiltrate the tumor (Figure 7D). The increased duration of pan-tumor T cell activation prior to exhaustion induces the observed differences in tumor volume.

### 3.6. Impact of T Cell Activation

The impact of cytokine-mediated delays in T cell exhaustion on tumor volume, as well as the common combination of bacterial therapies with immune checkpoint blockades, leads us to investigate the role of T cell exhaustion and activation in our model system. T cell activation is modulated by changing the activation constants *t*_1_ and *t*_2_. If there is no T cell activation present in the system, then the addition of bacterial therapies is insufficient to strongly impact tumor growth. Increasing bacterial density up to *S_B_*_,0_ = 10^6^ CFU results in a minimal alteration of the tumor volume curve (Figure 8A). Pan-tumor T cells have a greater impact on tumor cell number than bacteria antigen-specific T cells (Figure 8B). Pan-tumor T cells kill all cancer cells as opposed to only those that have been invaded by bacteria, which leads to this difference. However, bacteria antigen-recognizing T cells are critical for inducing the immune cell death cascade.

Bacterial therapies are often combined with checkpoint blockades such as PD-L1 antibodies, which limits CTL exhaustion, enhancing treatment efficacy [77,78]. We model the co-administration of the immune checkpoint blockade by minimizing the rate of T cell exhaustion, *d*_3_. Decreasing the rate of T cell exhaustion from 0.105 (light grey) to 0.001 (black) day^−1^ dramatically decreases tumor growth, leading to complete tumor regression (Figure 8C). Preventing T cell exhaustion leads to greater numbers of T cells that do not decrease over the 30-day treatment period (Figure 8D,E).

## 4. Discussion

*Salmonella* therapies, and bacterial therapies in general, have demonstrated strong potential in preclinical studies, but have not had the same efficacy in clinical trials [1,6]. Modeling systems can be critical to understanding which parameters need to be improved for therapeutic efficacy. Here we develop a model that describes *Salmonella* delivery of an exogenous antigen to cancer cells [5]. *Salmonella* invasion sensitizes tumor cells for destruction by one particular subset of T cells, *E_B_*, while also stimulating cytokine production in the tumor microenvironment. We assume that all tumor cells display some form of neoantigen that can be recognized by anti-tumor T cells, *E_T_*, which are stimulated by interactions with cancer cells with support from inflammatory cytokines. Using this model system, we observe several key results. The first is that interplay between extracellular and intracellular *Salmonella* is critical for reducing tumor volume. While intracellular *Salmonella* reduces initial tumor growth (Figure 3C,D), a reserve of extracellular *Salmonella* prolongs the population of invaded cancer cells, reducing tumor growth over time (Figure 6E–G). The second is that immune activation is critical for therapeutic success.

Comparison of our model to published in vivo data suggests model accuracy (Appendix A). We compare our results to data, with permission, from the article “Intracellular *Salmonella* delivery of an exogenous immunization antigen refocuses CD8 T cells against cancer cells, eliminates pancreatic tumors and forms anti-tumor immunity” by Raman et al. [5]. The data from this article is selected due to the release of specific antigens into cancerous cells and the presence of antigen-specific T cells. Notably this differs from the proposed modeling results, as a specific antigen is released rather than general bacterial antigens. Comparing tumor growth in vivo between saline and bacterial injections (Appendix A) to modeling results (Appendix A, adapted from Figure 5D) shows similar results with saline-injected tumors, reaching maximal volume by day 20, and *Salmonella*-treated tumors at half-maximal volume by day 30. In results based on the presence or absence of bacterial specific T cells in vivo (Appendix A) and in modeling (Appendix A, adapted from Figure 8B), we observe more rapid tumor growth without bacteria-specific T cells. Together these qualitative comparisons support confidence in our model.

Using PRCC analysis we are able to determine which rate constants in our model have the greatest impact on tumor growth after treatment with *Salmonella* (Figure 2B). PRCC analysis finds that, in addition to the rate of tumor growth, factors such as *Salmonella* invasion into the tumor, *k*_1_; *Salmonella* intracellular invasion, *k*_2_; T cell cytotoxicity terms, *s*_1_ and *h*_1_; and the influx of naïve T cells into the system significantly impact tumor growth. It is also interesting to note which factors are not significant in characterizing untreated tumor growth and which become significant after treatment with *Salmonella* (Figure 2A,B). Treatment with *Salmonella* increases the significance of both *Salmonella* specific kinetic terms, as well as T cell specific terms, indicating that both serve parallel roles in tumor treatment. In a bacterially treated system, the cytotoxicity rate of non-bacterially specific T cells, *s*_1_, significantly impacts tumor growth; however, the cytotoxicity rate of bacterial antigen-specific T cells, *s*_2_, does not. This suggests that lysis of bacterially invaded cancer cells is an inciting event, but not a strong enough effect to completely clear tumors. This matches with our model suggesting that only a fraction of the tumor cells are invaded by *Salmonella* (Figure 3D).

Increasing the *Salmonella* presence in tumors reduces tumor growth. In Figure 5 we observe that regardless of tumor size, increasing the delivered quantity of *Salmonella* results in smaller tumors 30 days after injection. The ratio of injected *Salmonella* to tumor cells does not explain the anti-tumor efficacy. With a small tumor (10^5^ cells) a *Salmonella*/cancer cell ratio of 1:10 (Figure 5A, yellow) results in initial tumor control. However, in large tumors (10^7^ cells), the same ratio has a minimal effect on the cell population (Figure 5G; purple). The initial volume of the tumor has a greater impact on cancer cell proliferation. The *Salmonella* dose cannot be increased indefinitely as there are limitations to the number of *Salmonella* that can be safely injected. When *Salmonella* are injected intratumorally rather than intravenously, there is a significant reduction in tumor volume (Figure 3). Finally, when the rate of *Salmonella* infiltration into the tumor, *k*_1_, is increased, tumor volumes are reduced (Figure 6A). Interestingly, according to our model, at most only 35 percent of the initial volume of cancer cells are invaded with *Salmonella* (Figure 3C). When injected intravenously, this number drops to only ten percent of cells (Figure 3D). This suggests that direct bacterial cell death, *k*_3_, is not the primary cause of tumor clearance, which is supported by the PRCC results (Figure 2B). This paints a relatively simple picture where increasing *Salmonella* numbers reduces tumor growth. This is not surprising as *Salmonella* stimulate cytokine production, directly lyse cancer cells, and make cells they have invaded targets for T cell mediated cell death.

Increasing intracellular invasion, *k*_2_, does not control tumor growth. This result is somewhat counterintuitive. Increasing intracellular invasion initially reduces tumor growth (Figure 6F), but this effect is short-lived and, after 15 days, tumors with the slowest rates of intracellular invasion have the smallest tumors (Figure 6E). The PRCC analysis suggests that increasing *k*_2_ would decrease tumor volumes at day 15, which matches Figure 6. Increasing invasion rates shifts the peak of invaded cancer cells earlier, with a sharp decrease after the *Salmonella* have cleared from the blood (Figure 6G). Lower rates of intracellular invasion maintain the extracellular compartment of *Salmonella*, which then invade cancer cells over time, creating a long tail of invaded cancer cells. An early peak in invaded cancer cells decreases the peak of bacterial antigen detecting T cells (Figure 6H). Expanding the antigen presentation window by increasing the duration of invaded cancer cells in the tumor microenvironment increases the activation of T cells targeting invaded cells. This can also be observed in the intravenous versus intratumoral injection case, where intravenous injection results in more bacterial T cells (Figure 3G), and in the investigation of changing the rate of intratumoral infiltration, where decreasing the rate of intratumoral invasion, *k*_1_, results in longer tails in the number of invaded cancer cells (Figure 6B) and increases in bacterial T cells (Figure 6C). Further experimentation is necessary to determine if this mechanism is maintained in vivo.

Immune activation is critical for *Salmonella*-based immunotherapies. *Salmonella* injections in systems without T cell activation demonstrate minimal efficacy (Figure 8A), and nonspecific pan-tumor immunity is critical for tumor reduction (Figure 8B). With the treatment of 10^6^ CFU of *Salmonella*, hot tumors have a stronger response than cold tumors (Figure 4C,D). Increasing the rate of cytokine production in response to *Salmonella* reduces tumor growth overtime (Figure 7B), although the impact on tumor growth is mostly observed after the majority of the cytokines have cleared (Figure 7A). Increased cytokine production has a minimal effect on the initial bacterial T cells (Figure 7C) but does lead to increased numbers and longer residence of pan-tumor T cells (Figure 7D). Increased extracellular *Salmonella* increases cytokine levels (Figure 6J). Bacterial therapies are often combined with checkpoint blockades to limit T cell exhaustion. Reducing the T cell exhaustion rate to model co-treatment with checkpoint inhibitors results in tumor clearance and long-term activation of T cells (Figure 8C–E). Even in cold tumors that do not respond to ICI as a monotherapy, the model suggests it should be added in combination with most bacterial therapies, which benefit greatly when T cells are less exhausted and clear slower from tumors. As the model suggests that bacterial therapies are more effective in environments with activated T cells, this raises a question of therapeutic efficacy in cold tumors. *Salmonella* are known to increase immune activation that could overcome immunosuppression, but this effect must be characterized in vivo.

Models are estimations of biological systems that make assumptions and simplifications of biological reality. This model has several limitations that must be considered when using the results to guide therapeutic design. First, this model is an ODE model and does not take into account the spatial dynamics within the tumor. Tumor heterogeneity is one of the major challenges with cancer therapy today. While *Salmonella* are known to proliferate throughout the tumor, including away from vasculature, this heterogeneity limits immune cell infiltration, which can result in tumor immune escape. ODE models provide system-level insights but are limited in predicting direct cell–cell interactions. This is also a stochastic model that relies on parameter estimation. Some parameter estimations are performed based on in vitro results, such as bacterial cytotoxicity and cytokine production. The complex tumor microenvironment and increased cell types can modulate these parameters in vivo. Finally, this model engages with cytotoxic T cells as the immune response. Tumors have a wide variety of immune subtypes, including macrophages, dendritic cells, helper T cells, and NK cells, which are not captured in this model. The results of this model should be used to guide future in vivo experimental design but not to replace further experimentation.

*Salmonella* therapies rely on the interplay between direct cytotoxicity and immunogenic cell death to generate an anti-tumor microenvironment. Our results suggest that future bacterial therapies should focus less on direct cytotoxicity and more on bystander effects, including immune stimulation and pro-drug activation as their mechanism of action. An optimal strategy may include having a subset of the *Salmonella* remain outside of cancer cells to produce an immune-active microenvironment, while another set invades cancer cells to initiate T cell-mediated cell death by memory T cells. Cytokine production in response to *Salmonella* reduces tumor growth (Figure 7). This suggests that a promising strategy would be to have extracellular *Salmonella* producing cytokines or PAMPs to stimulate immune activation and transition the microenvironment into a “hot” one. In clinical systems, the potential for a cytokine storm is a major risk that must be considered. In practice, modeling recommendations must be balanced with potential toxicity. Long-term bacterial presence within the tumor and consistent intracellular invasion may be more effective than a bolus of intracellular bacteria, providing long-term stalling of tumor cell proliferation and exposure to bacterial antigens. Improved knowledge of the interaction of bacteria with tumors and the immune cells within them would improve cancer care by allowing for more thoughtful design of targeted therapies.

## Figures and Tables

**Figure 1 bioengineering-12-00751-f001:**
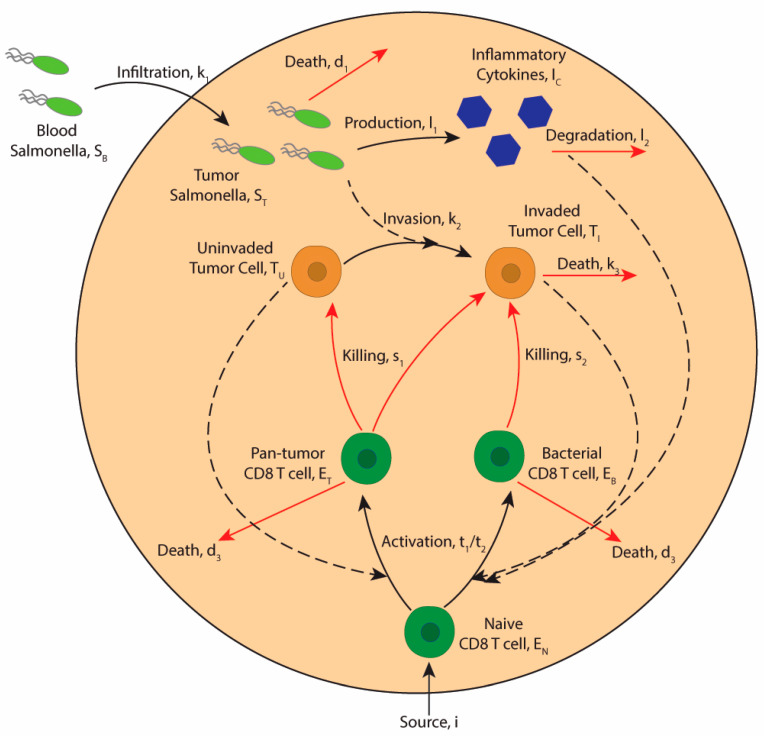
Mechanisms of bacterial immune therapy. *Salmonella* from the blood (*S_B_*) infiltrate tumors (*S_T_*). Once inside the tumor microenvironment, these *Salmonella* invade cancer cells (*T_U_*), causing them to become infected and display bacterially delivered antigens (*T_I_*). Intracellular *Salmonella* directly induce cancer cell death (*k*_3_) and prevent tumor cell division. Intratumoral *Salmonella* induce the production of immunostimulatory cytokines (*I_C_*), which assists in T cell activation. Without pro-inflammatory cytokines, T cell activation is dampened. Naïve T cells enter the tumor microenvironment from circulating T cells (*i*). The interaction between naïve T cells (*E_N_*) and invaded cancer cells leads to the activation of bacterial antigen-specific T cells (*E_B_*). The interaction of all tumor cells (*T_U_ + T_I_*) stimulates the activation of pan-tumor T cells (*E_T_*). Bacteria-specific T cells kill infected cancer cells by immunogenic cell death. Pan-tumor T cells target and kill both invaded and uninvaded cancer cells.

**Figure 2 bioengineering-12-00751-f002:**
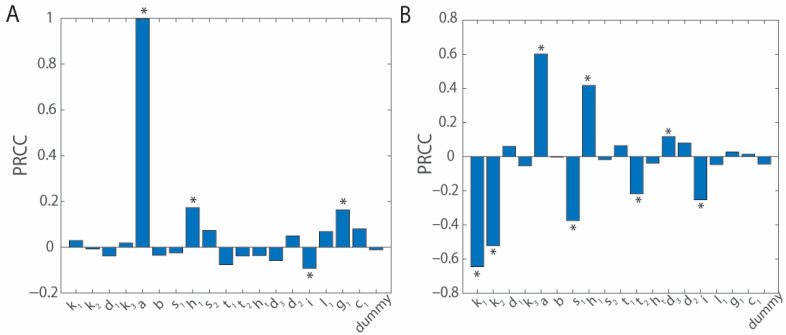
Sensitivity analysis. PRCC analysis of the model was conducted with 500 runs varying the parameters by up to one order of magnitude in each direction. The starting values for the parameters and initial conditions are as described in Table 1. Total tumor volumes (*T_U_ + T_I_*) were evaluated at 15 days post-bacterial injection for significant differences. Sensitivity was modeled with (**A**) *S_B_*_,0_ = 0 and (**B**) *S_B_*_,0_ = 10^6^ CFU (* indicates *p* < 0.05 for the PRCC values).

**Figure 3 bioengineering-12-00751-f003:**
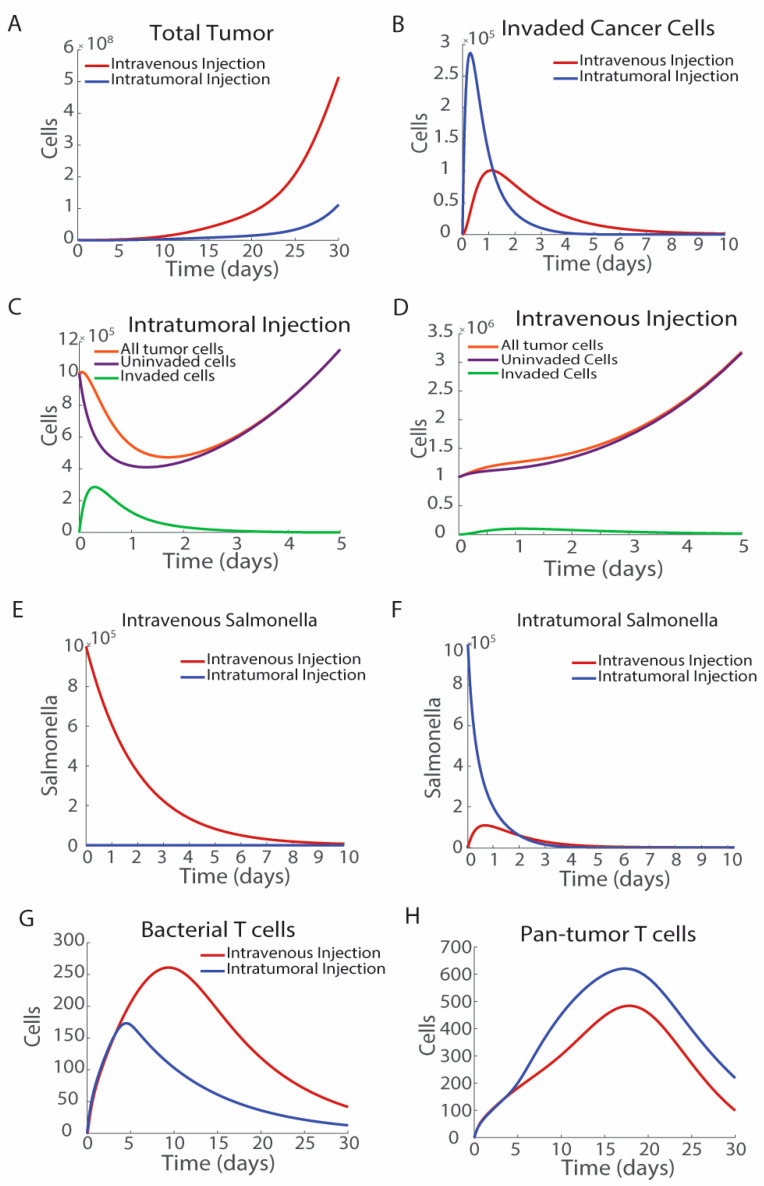
Effects of route of administration. Intratumoral injection (blue) of bacteria improves therapeutic efficacy over intravenous injection (red). (**A**) Intratumoral injection delays increases in tumor volume compared with intravenous injection. (**B**) Intratumorally injected *Salmonella* result in more invaded cancer cells and a rapid peak as opposed to intravenously injected *Salmonella*, which has a smaller flatter invaded cell curve. (**C**,**D**) The distribution of invaded cancer cells (T_U_; green), uninvaded cancer cells (T_I_; purple), and the total number of cancer cells (T_U_ + T_I_; orange) after injection with intratumoral (**C**) and intravenous (**D**) *Salmonella*. After intravenous injection, *Salmonella* migrate from the blood to the extracellular tumor space to inside cancer cells. (**E**) Bacterial density in the blood decreases over time, with the majority of the bacteria cleared 5 days post-injection after intravenous injection. After intratumoral injection there are no *Salmonella* in the blood. (**F**) After intravenous injection, intratumoral *Salmonella* increase, reaching a peak one-day post-injection, with a little over one tenth of the bacteria in the tumor microenvironment. After intratumoral injection, bacteria decrease rapidly. (**G**) Intravenous injection leads to more bacteria-specific T cells but (**H**) less pan-tumor T cells.

**Figure 4 bioengineering-12-00751-f004:**
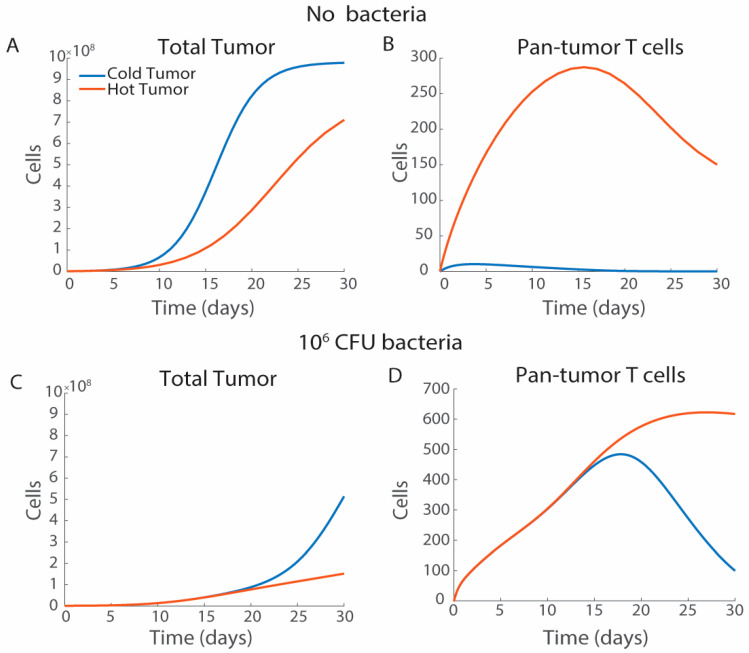
The effect of hot and cold tumors. The model was evaluated for response in immunogenically cold (blue, *I_C_*_,0_ = 10, *l*_2_ = 0.5137) and immunogenically hot (orange, *I_C_*_,0_ = 100, *l*_2_ = 0) conditions. (**A**) Without bacterial treatment (*S_B_*_,0_ = 0), cold tumors reached carrying capacity 22 days after model start, while hot tumors did not reach carrying capacity before 30 days. (**B**) In immunogenically cold tumors there was minimal activation of tumor antigen T cells, while there was significant T cell activation in immunogenically hot tumors. (**C**) Tumors treated with *Salmonella* (*S_B_*_,0_ = 10^6^ CFU) showed delayed cell proliferation compared with the untreated tumors. Bacterial administration to hot tumors (orange) remained below 2 × 10^8^ tumor cells 30 days post-simulation start. (**D**) With *Salmonella*, immunogenically cold tumors decreased the number of tumor T cells over time, while in hot tumors this decrease was not present.

**Figure 5 bioengineering-12-00751-f005:**
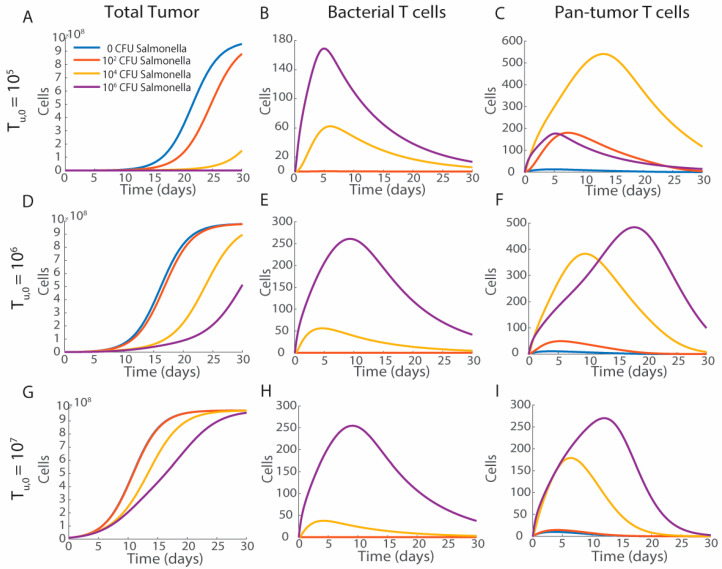
Increasing the number of bacteria delivered at the start of treatment decreases tumor volume over time. Increasing initial concentrations of *Salmonella*; 0 CFU (blue), 10^2^ CFU (orange), 10^4^ CFU (yellow), to 10^6^ CFU (purple); (**A**,**D**,**G**) slowed tumor growth (**B**,**E**,**H**) increased the formation of bacterial antigen-specific T cells and (**C**,**F**,**I**) pan-tumor T cells over the course of thirty days. The initial tumor volume impacted treatment efficacy. (**A**–**C**) Bacterial treatment of small initial tumors T_U,0_ = 10^5^ cells delayed growth and a dose of 10^6^ CFU of *Salmonella* resulting in tumor clearance (purple). (**D**–**F**) A starting tumor volume of 10^6^ cells had the widest range of tumor responses without tumor clearance**.** (**G**–**I**) A starting tumor volume of 10^7^ cells reduced the efficacy of *Salmonella* treatment and all tumors grew rapidly.

**Figure 6 bioengineering-12-00751-f006:**
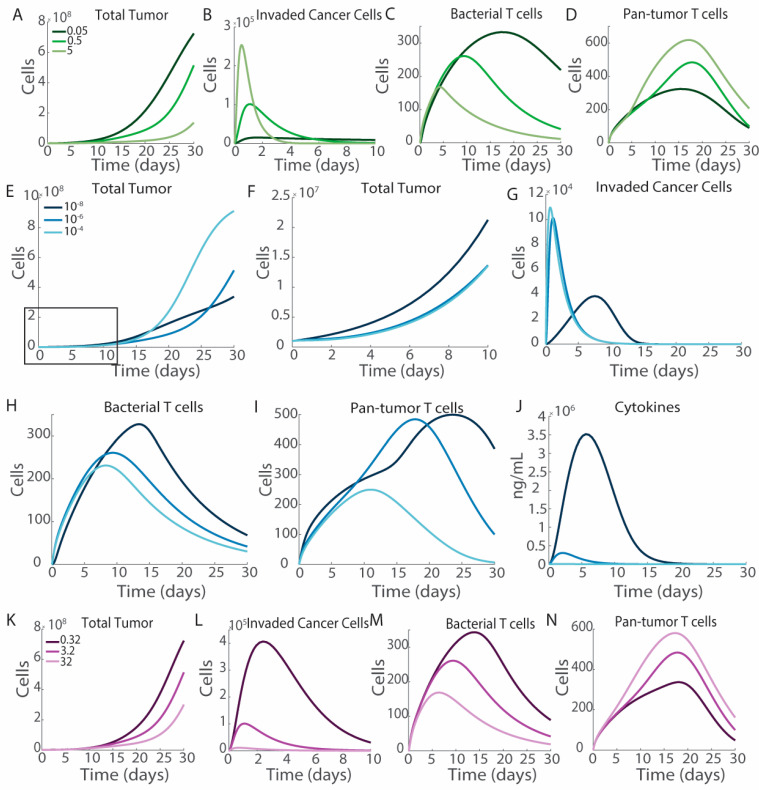
Modifying the bacteria impacts therapeutic efficacy. (**A**–**D**) Changing the rate of tumor infiltration, *k*_1_, from 0.05 day^−1^ (dark green), 0.5 day^−1^ (medium green), to 5 day^−1^ (pale green). (**A**) Increasing the rate of bacteria entering the tumor reduces tumor volume. (**B**) Increasing *k*_1_ leads to more invaded cancer cells that clear faster. (**C**) Increasing *k*_1_ decreases the number of bacterial T cells, while (**D**) increases the number of pan-tumor T cells. (**E**–**J**) Changing the rate of intracellular invasion, *k*_2_, from 10^−8^ cells day^−1^ (dark blue), 10^−6^ cells day^−1^ (medium blue), 10^−4^ cells day^−1^ (light blue) increases tumor growth. (**E**,**F**) In the first 10 days after treatment, increasing the invasion rate delays tumor growth; however, after 15 days increasing rates of intracellular invasion leads to rapid tumor cell expansion. (**F**) is an increased view on the first 10 days of (**E**). (**G**) Slow intracellular invasion (dark) leads to fewer invaded cancer cells that peak in number 7 days after bacterial injection rather than 1 day as with faster invasion (medium and light). (**H**,**I**) Slower tumor invasion leads to increased T cells. (**J**) Slow intracellular invasion increases cytokine production in tumors. (**K**–**M**) Increasing the rate of direct bacterially mediated cancer cell death, *k*_3_, from 0.32 (dark purple), 3.2 (medium purple), to 32 (light purple) improves therapeutic efficacy. (**K**) Increasing bacterial cell death reduces tumor growth and (**L**) decreases the overall number of invaded cancer cells. (**M**) Increasing *k*_3_ decreases the number of bacterial T cells while (**N**) increasing pan-tumor T cells.

**Figure 7 bioengineering-12-00751-f007:**
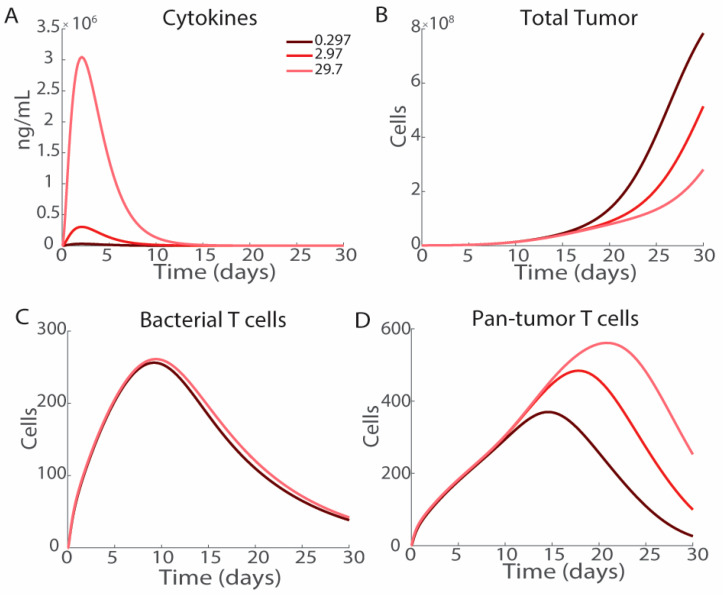
Effect of cytokine production. Increasing the rate of cytokine production from 0.297 (dark red) to 2.97 (medium red) and 29.7 (light red) improves therapeutic efficacy. (**A**) Increasing the rate of cytokine production by bacteria produces more cytokines early in tumor treatment. The majority of the produced cytokines are cleared 15 days post-injection. (**B**) Increased cytokine production by bacteria reduces tumor growth after the first 15 days of treatment. (**C**) Cytokine production has a minimal effect on bacterial T cells. (**D**) Cytokine production increases the number of pan-tumor T cells after 15 days.

**Figure 8 bioengineering-12-00751-f008:**
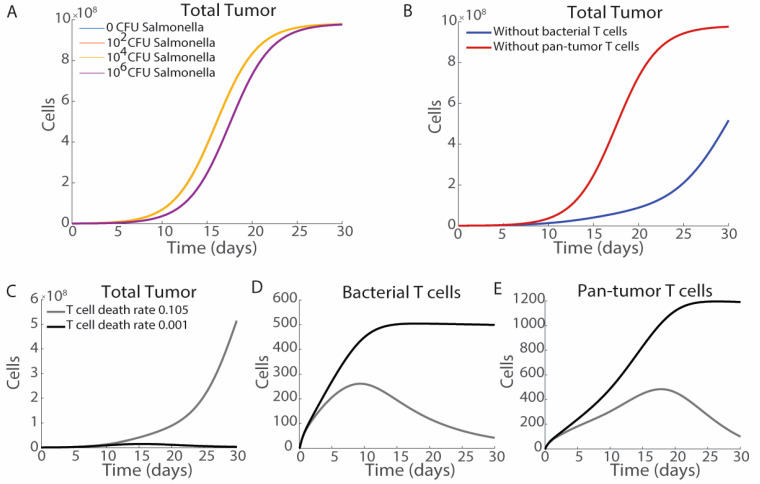
*Salmonella* therapies are dependent on the T cell dynamics within the tumor. (**A**) Without T cell activation there is minimal change in tumor growth until delivery of 10^6^ CFU of *Salmonella*, which only provides a minor delay in proliferation. (**B**) Pan-tumor T cells are critical for controlling tumor cells even with bacterial treatment (10^6^ CFU). Without bacterial T cells (blue) there is still a strong reduction in tumor volume. Without pan-tumor T cells (red) the tumors still reach carrying capacity 22 days post-infection. Checkpoint blockade therapies slow the rate of T cell exhaustion (**C**–**E**). (**C**) Reducing the T cell exhaustion rate from 0.105 (light grey) to 0.001 (black) results in tumor clearance. (**D**,**E**) Reducing T cell exhaustion leads to more bacterial and pan-tumor T cells, which remain at high density rather than clearing.

## Data Availability

The model may be provided by reasonable request to the authors or by request on GitHub: https://github.com/lmhowell/BacterialModeling.

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
