# Peer review of "Mathematical Modeling of *Salmonella* Cancer Therapies Demonstrates the Necessity of Both Bacterial Cytotoxicity and Immune Activation"

_bioengineering, 2025, doi:10.3390/bioengineering12070751_

Round 1

Reviewer 1 Report

Comments and Suggestions for Authors

The manuscript describes the numerical modelling of the effect of an anticancer agent on tumor growth. Such simulation models are a very valuable route towards visualising, understanding and ultimately optimising the mode of action of active agents in a multistage cascade. In general, the study is well conceived and presented. There are however, some issues which should be considered to improve the quality of the presentation;

(i) Supporting references should be provided to support statements in several places;

"Bacterial therapies are a promising new cancer immunotherapy with multiple clinical 28 trials underway."

(ii) There are a few places where the text should be checked for grammar - one example is;

"One way that Salmonella direct T cell cytotoxicity to tumor cells by delivery of antigen"

"Bacterial therapies do not depend on cytotoxicity alone. Effective Salmonella thera-60 pies increase the immunogenic tumor microenvironment and rely on the immune system for tumor clearance."

"Mathematical models of cancer progression and treatment have been critical to understanding mechanisms undergirding immunotherapies and designing more effective therapies."

"Parameters from commonly cited cancer immunotherapy models were chosen for communicability between this model and other models in the field."

"Hot tumors have reduced tumor growth and increased T cell activation compared to 347 cold tumors"

(iii) The authors should indicate whether the model will be made available Open Access

(iv) In Figure 1, what is "Source,i"

(v) Equation 1 implies that there is no other route for clearance of salmonella, and that all of a dose will eventually enter the tumor. It might be worth including an alternative clearance (even natural blood flow), as the relative rates may influence the downstream considerations.

(vi) The authors should comment on the  relative ratios of salmonella doses and number of tumor cells.

(vii) In Sections 2.21,2,3,4 more frequent/direct reference to the Equations and the relevant parameters should be made, so the reader can better understand the construction of the model.

(viii) As the manuscript progresses, reference is increasingly made to "tumor size", "tumor volume", "tumor growth rate". None of these are explicitly modelled, and so the authors should explicitly clarify on what basis they are inferred. 

(ix) The 2.4. Sensitivity analysis indicate that the model is highly sensitive to certain parameters, and in particular on the parameter a. either at this point or in the discussion, the authors  should comment again on their method of estimating, and the potential variability of these most sensitive parameters.

(x) "Sensitivity analysis showed that the rates of bacterial infiltration into tumors and intracellular invasion critically impacted tumor volume," -  Is this really demonstrated? Or does the sensitivity analysis indicate that the model is very sensitive to these parameters?

(xi) Figure 2 - the y-axis should be labelled.

(xii) "The rates of Salmonella invasion into the tumor k1 (p =  3.6213x10-60) and cancer cells k2 (p = 3.094x10-36) are negatively correlated with tumor volume." Where is this shown, given that tumor volume is not explicitly calculated?

(xiii) "3.1. The route of bacterial administration impacts tumor growth" The first part of this section is not Results, and should be included in the Methods section.

(xiv) "With this initial dose of bacteria, fewer than one tenth of the total volume of cancer cells are invaded by Salmonella" -  This will depend on (vi) the relative ratio of the dose to target cells.

Author Response

We thank the reviewer for their comments. Our responses are discussed below.

  1. Clinical trial numbers were added to support this reference
  2. These sentences were edited for clarity and may be found on lines 43, 63, 81, 229, 374, in the tracked changes version of the revisions.
  3. The code for this model has been uploaded to GitHub. Currently it is in a private repository and access can be granted upon request to the authors or user lmhowell on github.
  4. Source, i is the source of naïve T cells from circulation and a clarification on this point was added to the figure caption
  5. Adding an additional clearance mechanism was tested and did not impact tumor infiltration or final volume until the rate of clearance vastly exceeded (5 day-1) what has been seen in vivo and has been added as a supplemental figure
  6. Added some discussion on Salmonella ratios in the discussion on lines 611-617.
  7. Additional pointers were added to these sections.
  8. Tumor volume/size is assumed to be proportional to the number of liver tumor cells (TU + TI) and has been clarified in section 2.2.2. Tumor growth is defined as the proliferation of tumor cells and has been clarified as well. “Tumor volume is assumed to be proportional to the number of live tumor cells, (TU + TI). Tumor growth is the change in tumor cell number over time.”
  9. Added some description of PRCC “Briefly, each parameter of the model was randomly assigned a value within an order of magnitude of the literature determined values. The model was run using these parameters. The total tumor cell number, TU+TI, 15 days after bacterial injection was plotted against the randomly generated values for each parameter to determine their impact on tumor cell count. For tumor growth rate, a, and carrying capacity, b, parameters were chosen within a factor of two as these values were more influential than other factors on final tumor cell count.” (2.4)
  10. We edited this section to better reflect that sensitivity analysis is a measure of the model rather than the biological mechanism (line 284)
  11. Axis labels added (PRCC)
  12. We changed this to refer to cell number to more accurately reflect the model results (line 305)
  13. Respectfully, we disagree that the method of bacterial administration is not a result and belongs in the methods section. Routes of therapeutic delivery are an active question in the field and a discussion of this belongs in the results section.
  14. Changed to reflect that is dependent both on initial dose and tumor volume (line 360).

Reviewer 2 Report

Comments and Suggestions for Authors

This is a very interesting paper modelling different parameters of immune response both to tumor and to Salmonella. I do not have major remarks. Some minor ones are shown in pdf file. Please check Fig. 6 legend, may be some figures will be more understandable and comparable if the ordinate scales are the same, some minor errors.   

Author Response

6 and figure 2. Figure 3E,F were edited to included both intravenous and intratumoral injection to clarify the distinction between delivery method and bacteria location.

In figure 3 we made the axes the same in E-F. However, for C-D the axes were left as is as the purpose of the figures is to highlight the relationship between invaded and uninvaded cells within a treatment type. The comparison of intravenous versus intratumoral injections can be observed in 3B. The AUC analysis was added in figure 3B. “Overall, both methods of bacterial injection resulted in similar numbers of invaded cancer cells during the first 10 days. The area under the curve in intravenously injected Salmonella is 2.9963x105 while that of intratumorally injected Salmonella is 3.0694x105, indicating that it is the timing of cell invasion rather than the total number of invaded cells that impacts efficacy (Figure 3B).” Thank you for pointing out this analysis method as it added an interesting discussion point.

The sentence in the abstract was edited for clarity.

Reviewer 3 Report

Comments and Suggestions for Authors

The manuscript entitled "Mathematical modeling of Salmonella cancer therapies demonstrates the necessity of both bacterial cytotoxicity and immune activation" presents an ordinary differential equation (ODE)-based mathematical model integrating engineered Salmonella bacterial therapies with classical anti-tumor immunotherapies. The model aims to elucidate how different bacterial properties (tumor infiltration rate, intracellular invasion, cytotoxicity, and cytokine production) and immune parameters (T-cell activation, cytokine dynamics, checkpoint blockade) contribute to tumor clearance. Using sensitivity analysis and various simulation scenarios, the authors identify key determinants of therapeutic efficacy, providing guidance for optimizing bacterial immunotherapy designs.

- The model is sophisticated and biologically insightful; however, its clinical translatability is not adequately validated. There is no comparison of model predictions with in vivo or clinical data beyond referencing prior work (e.g., [5], [6]). While parameter estimation uses data from the literature, direct validation would greatly strengthen the confidence in the model's predictions. Authors should consider including at least a qualitative or semi-quantitative comparison to published in vivo data.

-Many parameters were derived from in vitro studies or previously published models. For example, invasion rates (k1, k2), cytotoxicity rates (k3, s1, s2), and cytokine production rates (l1) may differ substantially in the complex tumor microenvironment. Authors should discuss more thoroughly the uncertainty associated with these parameters and possibly conduct uncertainty analysis beyond PRCC (e.g., Monte Carlo simulations).

-One of the most striking findings is that increasing intracellular invasion (k2) paradoxically results in higher tumor volumes at later time points. While this is an interesting model outcome, biologically it is somewhat counterintuitive. The underlying mechanistic explanation (shortened antigen presentation window, T cell dynamics) needs to be better elaborated, and ideally supported by empirical data if available. This point should be more critically discussed in the Discussion.

-The model suggests that Salmonella therapies are largely ineffective without pre-existing or induced T-cell activation. This raises the question of how such bacterial therapies would perform in highly immunosuppressive (“cold”) tumors, which are often the target population for novel immunotherapies. While the authors partially address this in section 3.2 (Lines 346-384), a deeper discussion is warranted, particularly with respect to clinical implications.

-The model does not account for spatial heterogeneity (e.g., variable oxygenation, vascularization, immune cell infiltration), which are known to critically influence bacterial colonization and immune activity. The authors mention prior PDE models (Lines 81-83), but do not justify sufficiently why spatial aspects are neglected here. While ODE models are valuable for system-level insights, limitations due to lack of spatial resolution should be acknowledged more explicitly.

-While increasing cytokine production (l1) improves therapy efficacy in the model, real-world toxicity of systemic cytokine elevation (e.g., IL-6, TNFα storms) is a major clinical barrier. Authors should caution that such model-based predictions must be carefully balanced against toxicity risks in practice.

-The manuscript would benefit from a dedicated section discussing model limitations (e.g., lack of spatial dynamics, stochasticity, parameter uncertainty, simplification of T cell compartments), to guide readers on how to interpret the findings.

Author Response

We thank the reviewer for their comments on this manuscript.

It is true that the parameters are derived from in vitro results and thus have limitations. These limitations are discussed in the discussion (lines 673-677) and in section 2.3 (lines 208-210). PRCC is a useful method for dealing with the uncertainty of these parameters as all parameters are varied in each run, providing an system view of the model variability.

It is true that the result of increasing intracellular invasion is surprising. We expanded our discussion of hypothetical mechanisms in the discussion (lines 628-646) and acknowledge that in vivo experimentation is necessary to determine the validity of this hypothesis. The purpose of this model is not to confirm results, but to provide hypothesis for testing in the future.

We added a paragraph in the discussion on the model limitations to better inform the reader (lines 666- 681). In this paragraph we addressed the reviewers concerns on spatial heterogeneity, parameter uncertainty, and limited immune cell types. We also discussed the safety concerns with increasing cytokine production in the discussion (lines 691-698). We also discussed the potential limits of bacterial therapies in cold tumors in the discussion (lines 662-665).

To compare our model to prior work we performed qualitative analysis on tumor growth curves to published in vivo murine tumor growth data from the paper “Intracellular Salmonella delivery of an exogenous immunization antigen refocuses CD8 T cells against cancer cells, eliminates pancreatic tumors and forms antitumor immunity” by Raman et al. These comparisons may be found in the discussion on lines 579-593 and in Supplemental Figure 2.

We thank the reviewer for expanding our discussion on the limitations of modeling and the importance of discussing the need for clinical translation in interpreting the results.

Round 2

Reviewer 3 Report

Comments and Suggestions for Authors

From my side, it can be publisable now.